# Implementation fidelity in leprosy care and support for disability prevention and management in Rupandehi, Nepal: A qualitative study

Sudip Nepal[1,2*], Ari Probandari[3], Amit Timilsina[1], Anisha Shrestha[1], Prakash Chandra Joshi[4], Riris Ahmad Andono[5]

1 Research and Community Development Center, Kathmandu, Nepal, 2 Faculty of Medicine, Public Health and Nursing, Universitas Gadjah Mada, Yogyakarta, Indonesia, 3 Department of Public Health, Faculty of Medicine, Universitas Sebelas Maret, Surakarta, Indonesia, 4 James P. Grant School of Public Health, BRAC University, Dhaka, Bangladesh, 5 Centre for Tropical Medicine, Public Health and Nursing, Universitas Gadjah Mada, Yogyakarta, Indonesia

* nepalsudip25@gmail.com

## Abstract

### Background

Implementation fidelity is critical for the efficient delivery of health services including leprosy services. Healthcare providers are important in monitoring the disease's progression, managing complications, and cross-checking prescribed medications. This study explored implementation fidelity in leprosy care and support for disability prevention and management in Rupandehi district, Nepal.

### Methodology

A qualitative case study design was adopted based on implementation research principles. From 25th February to 30th April 2024, data were collected through multiple sources and methods, including key informant interviews, focus group discussions, and observation. Semi-structured interview guidelines and qualitative checklists facilitated the data collection process. Participants were chosen using purposive and selective sampling methods. The data were inductively coded using qualitative analysis software. Thematic analysis was done with codes generated and aggregated to form sub-themes and develop themes.

### Results

The study revealed that healthcare providers consistently adhered to national leprosy operational guidelines, ensuring sufficient fidelity by prompt multi-drug therapy, case diagnosis, complicated case referral, and regular follow-up. In contrast, poor adherence was demonstrated in the complication management of lepra reactions, ulcer

which permits unrestricted use, distribution, and reproduction in any medium, provided the original author and source are credited.

**Data availability statement:** The minimal data set underlying the findings of this study is provided within the manuscript and its Supporting Information files. This includes the socio demographic characteristics of participants (Tables 1 and 2), comparison of national leprosy operational guideline and actual implementation of leprosy services (Table 3), excerpts (Results) and codebook (S1 Table), which are sufficient to replicate the findings of the qualitative analysis. Due to ethical concerns and participants confidentiality, full interview transcripts cannot be shared publicly. However, anonymized excerpts are included within the results section of manuscript to support the thematic findings. Additional anonymized data may be made available upon reasonable request to the corresponding author.

**Funding:** The primary author (Sudip Nepal) received funding support as a thesis grant for this study from the World Health Organization Tropical Disease Research (WHO-TDR) Special Postgraduate Programme on Implementation Research at Universitas Gadjah Mada (UGM), Indonesia. The funders had no role in study design, data collection and analysis, publication decisions, or manuscript preparation.

**Competing interests:** The authors have declared that no competing interests exist.

cases, and self-care. The major barriers to leprosy services were financial hardship, complication management, pill burden, drug side effects, and institutional obstacles. In contrast, the facilitators to leprosy services included adequate human resources, treatment supporter's involvement, effective communications, external development partner's role, transportation incentives, and local government support.

## Conclusions

Healthcare providers demonstrated sufficient adherence to leprosy operational guidelines. While significant gaps were evident in complication management, addressing financial and systematic barriers and leveraging facilitators is essential to strengthening leprosy care and support for disability prevention and management.

## Introduction

Leprosy, also known as Hansen's disease, has historically been one of the most stigmatised and significant public health issues in Nepal. The national leprosy elimination program has been implemented in Nepal since 1913/14, with ongoing government efforts to support people with leprosy [1]. Though Nepal achieved the milestone of eliminating leprosy as a public health problem in 2010, defined by the World Health Organization (WHO) as a registered prevalence of less than 1 case per 10,000 population. However, challenges such as early case diagnosis, complication management and disability prevention persist [1]. Leprosy causes long-term damage to peripheral nerves, skin, and other organs and develops complications during multi-drug therapy (MDT) treatment, such as leprosy reaction, neuropathic pain, and relapse and may lead to disability if not detected and managed promptly [2]. However, it is essential to note that leprosy is a disease, whereas disability is a potential consequence that may result from untreated or inadequately managed leprosy [2,3].

Leprosy remains a public health concern, especially in 23 globally prioritised countries, including Nepal [4,5]. As per data from 2023, In South Asia, India continues to report more than 10,000 new cases, while other countries (Bangladesh, Nepal and Sri Lanka) each reported 1000–10,000 new cases [5]. Although Nepal is geographically located in South Asia, early references often included it in Southeast Asia's statistics due to WHO regional classification with Nepal contributing to 2% of the total burden in South Asia [6]. As per WHO reports, globally, new cases decreased by 19.3% between 2013 and 2022. The COVID-19 pandemic reduced the detection of new cases due to restricted health service access and reduced active case finding [5]. For example, new case detection in Nepal dropped markedly in 2020 with the reporting of 2173 new cases but began to rebound in the post-pandemic period, with the country reporting 2,523 new cases in the fiscal year 2022/23 [1].

Rupandehi district has historically been one of the leprosy-endemic districts in Nepal. It still carries a notable leprosy burden, possibly underestimated due to multiple socio-health system factors. Rupandehi district has a registered prevalence of 1.2

per 10,000 population as of 2024. New case detection rates in Rupandehi have often been above the national average and grade 2 disability(G2D) cases still occur, indicating missed or late detections [1].

MDT is the WHO-recommended combination of antibiotics consisting of rifampicin, dapsone, and clofazimine administered over 6–12 months depending on the leprosy categorization to eliminate infection and prevent resistance [7]. The WHO classifies leprosy according to the WHO disability grading system, where grade 0 means no impairment, grade 1 means loss of sensation in the hand, eyes or foot, and grade 2 indicates a higher degree of disability compared to grade 1, G2D indicates visible deformities or damage in eyes, hands, or feet at diagnosis indicating more significant functional impairment and potential for long-term complications [8]. G2D was present in 189 (7.5%) new leprosy cases diagnosed [1]. This reflects the delay in diagnosis and indicates the complications such as nerve involvement, irreversible complications and the occurrence of new disabilities. This not only reflects the clinical burden but also increases stigma and social exclusion. These complications require comprehensive examinations including nerve function assessments and eye-hand-foot scores at the beginning and end of MDT treatment [4].

The healthcare system in Nepal can be categorised into three levels: primary (health posts), secondary (district hospitals), and tertiary (specialised institutions). Leprosy services are decentralised, with health posts serving as the primary point of contact for individuals seeking healthcare in the community. This initial contact serves as the first point for leprosy services such as leprosy diagnosis, MDT initiation, disability grading, complications management, referral and follow-up. The key health care providers responsible for implementing leprosy care and support services in the health post include the health post-in-charge and the leprosy focal person who has either a health assistant or auxiliary health worker background with leprosy training [9]. The level of health care provider's care and support required may vary depending on the individual's needs and the stage of the disease [4]. To guide the healthcare providers, Nepal has standard leprosy operational guidelines [10] adopted and aligned with WHO guidelines [11] to facilitate the implementation of care and support of leprosy for disability prevention and management in health facilities such as health posts, hospitals, and referral centres.

Despite these efforts, health posts in Rupandehi district lack essential skin smear tests, and skilled laboratory personnel to perform these tests. Consequently, they rely heavily on referrals. Participants from remote areas of Rupandehi district must travel 1–2 hours to reach Butwal for a skin smear test. Thus, healthcare providers refer patients to Butwal for these tests and to specialised hospitals in Kathmandu for leprosy-related complications, such as lepra reaction and ulcer management, if local health facilities cannot address them. Participants from these remote regions face considerable challenges due to transportation difficulties, often needing to travel 10–12 hours to reach specialised hospitals like Anandaban in Kathmandu for managing leprosy complications, resulting in out-of-pocket expenses, long waiting times and inconvenience for socio-economic disadvantaged populations.

In the socio-cultural context of Nepal, leprosy continues to carry a deep-rooted stigma, fueled by myths and misconceptions, which often end up in discrimination and social exclusion. Individuals may adopt harmful practices such as using inappropriate oils or herbal remedies, excessive soaking of wounds, or delayed wound care [12]. Such practices discourage seeking immediate leprosy care and support from healthcare providers, increasing the risk of complications and disabilities [13–15]. Good self-care practices like routine wound cleaning, protective footwear, and regular self-inspection can significantly lower the risk of ulceration and disability [10,11]. Engaging healthcare providers is crucial to recognise early symptoms, manage any side effects, and motivate individuals to practice good self-care to prevent disabilities and reduce stigma [16].

Implementation fidelity refers to the degree to which an intervention or program is delivered as intended by the program developers. High fidelity ensures that established leprosy recommendations and protocols are properly implemented by healthcare providers, resulting in better patient outcomes. In contrast, low fidelity explains why these underperform while being well-designed. Assessing implementation fidelity allows researchers to better understand how and why an intervention works, as well as the extent to which outcomes might be improved [17].

The relevance of studying implementation fidelity in leprosy care and support in the Rupandehi district lies in understanding how effectively Nepal's leprosy operational guidelines are implemented at the local level. Although national prevalence may be low, persistent leprosy complications, G2D cases, and stigma in leprosy-endemic districts like Rupandehi indicate gaps in policy and practice, justifying this research. While grey literature briefly discusses implementation fidelity in leprosy care and support [18], peer-reviewed studies systematically exploring fidelity in low-resource settings like Nepal remain scarce. This gap limits the development of evidence-based strategies aimed at enhancing service delivery [18].

This study explores the implementation fidelity of leprosy care and support for disability prevention and management in the leprosy-endemic region of Rupandehi, Nepal, which ultimately bridges the gap between existing leprosy policies and their practical application. Furthermore, this study provides empirical evidence, enabling healthcare providers and policymakers to understand the barriers and facilitators affecting implementation fidelity and suggest tailored interventions to enhance the quality of service delivery. In this study, data were collected once over two months, from 25th February to 30th April 2024, with participant enrolment beginning on 25th February 2024.

## Materials and methods

### Study setting

Nepal comprises 77 districts and 7 provinces, divided into three ecological belts: mountain, hilly, and terai belts. Rupandehi district is located in the terai belt of Lumbini province in southwestern Nepal and covers an area of 1,360 km$^2$. It is bordered by Palpa to the north (national border), India to the south (international border), Kapilvastu district to the west (national border), and Nawalparasi district (national border) to the east. Rupandehi is one of the leprosy-endemic districts of Nepal with a registered leprosy prevalence of 1.2 per 10,000 population as of 2024 [1].

### Study design

A qualitative case study design was adopted, grounded in the principles of implementation research. Data were collected through multiple sources and methods like key informant interviews (KIIs) with healthcare providers (n = 12), focus group discussions (FGDs) with self-help group members (SHGs) (n = 14), and qualitative observation at two high-caseload health facilities and a satellite clinic (n = 3) to ensure triangulation and enhance the credibility of the findings.

### Participant selection

Participants were selected via purposive and selective sampling, ensuring alignment with leprosy service availability and geographical locations. This allowed for a rich and in-depth exploration of the research question until data saturation. The data collection period for the study was from 25th February to 30th April 2024, with participant enrolment beginning on 25th February 2024.

For KIIs, ten health facilities from leprosy pocket areas were selected in consultation with the focal person for neglected tropical disease from the Health Office, Rupandehi. Twelve (12) KIIs were carried out with the healthcare providers, including the health facility in charge or the leprosy focal person, until data saturation. The health facilities selected for KIIs were Devdaha, Basantapur, Semara Bajar, Padsari, Tikuligadh, Bodhabar, Kerwani, Shankarnagar, Parroha and Dudhrakshya. In addition, two health facilities, Kerwani and Semara Bajar and a satellite clinic were observed qualitatively due to their higher caseloads and experience in managing complicated leprosy cases. FGDs were conducted with members of two SHGs, Namuna and Hariyali SHGs identified by the Health Office, Rupandehi. Participants over 18 years of age from the SHGs were contacted through The Leprosy Mission Nepal, Butwal. Fourteen (14) information-rich people with leprosy participants from SHGs were contacted and invited for the FGDs in two phases.

Table 1 summarizes the socio-demographic characteristics of 14 individuals with leprosy, all of them were associated with the SHGs. Nine (9) of the participants were males and 5 of them were female on the contrary. The age range varied

**Table 1. Socio-demographic characteristics of the people with leprosy.**

| S.N | Participant's code | Type of participant | Age | Occupation | Leprosy categorization | Gender |
|-----|-------------------|---------------------|-----|-----------|------------------------|--------|
| 1 | N1 | Individual with leprosy | 60 | Agriculture | Multibacillary (MB) | Male |
| 2 | N2 | Individual with leprosy | 50 | Agriculture | Multibacillary (MB) | Male |
| 3 | N3 | Individual with leprosy | 52 | Auto driver | Multibacillary (MB) | Male |
| 4 | N4 | Individual with leprosy | 45 | House manager | Multibacillary (MB) | Female |
| 5 | N5 | Individual with leprosy | 48 | House manager | Multibacillary (MB) | Female |
| 6 | N6 | Individual with leprosy | 25 | Job, cooperative | Multibacillary (MB) | Female |
| 7 | N7 | Individual with leprosy | 28 | Daily wage | Multibacillary (MB) | Female |
| 8 | H1 | Individual with leprosy | 48 | Shopkeeper | Multibacillary (MB) | Male |
| 9 | H2 | Individual with leprosy | 55 | Daily wage | Multibacillary (MB) | Male |
| 10 | H3 | Individual with leprosy | 50 | Daily Wage | Multibacillary (MB) | Female |
| 11 | H4 | Individual with leprosy | 60 | Shopkeeper | Multibacillary (MB) | Male |
| 12 | H5 | Individual with leprosy | 40 | Agriculture | Multibacillary (MB) | Male |
| 13 | H6 | Individual with leprosy | 25 | Shopkeeper | Multibacillary (MB) | Male |
| 14 | H7 | Individual with leprosy | 50 | Shopkeeper | Multibacillary (MB) | Male |

The WHO classification system is used to categorise leprosy as either PB or MB. MB cases have more than five skin lesions or positive skin smears, whereas PB cases have five or fewer lesions and negative skin smears [10,11].

from 25 to 60 years. The occupations of the participants varied from agriculture, daily wage, shopkeeper, house manager, auto driver and job holder at a cooperative.

Table 2 gives an outline of the socio-demographic characteristics of 12 healthcare providers, i.e., health facility in-charge or leprosy focal person from 10 health facilities, all associated with the health facilities of Rupandehi district. Seven (7) of the healthcare providers were males and 5 of them were female on the contrary. The age range varied from 29 to 48 years and professional experience varied from 6 to 25 years.

## Study tool and data collection

Prior appointments were made with healthcare providers for KIIs, people with leprosy for FGDs, and health facilities and satellite clinics for observations. An interview guide facilitated FGDs and KIIs, while an observation checklist supported the

**Table 2. Socio-demographic characteristics of the Healthcare Providers (HCP).**

| S.N | Code | Type of participant | Professional role | Age | Years of experience | Gender |
|-----|------|---------------------|-------------------|-----|---------------------|--------|
| 1 | KII 1 | Health Care Provider | Leprosy focal person | 33 | 10 | Male |
| 2 | KII 2 | Health Care Provider | Health facility In-charge | 34 | 6 | Female |
| 3 | KII 3 | Health Care Provider | Leprosy focal person | 48 | 25 | Female |
| 4 | KII 4 | Health Care Provider | Health facility In-charge | 42 | 20 | Male |
| 5 | KII 5 | Health Care Provider | Leprosy focal person | 29 | 6 | Female |
| 6 | KII 6 | Health Care Provider | Leprosy focal person | 40 | 9 | Female |
| 7 | KII 7 | Health Care Provider | Leprosy focal person | 33 | 7 | Female |
| 8 | KII 8 | Health Care Provider | Leprosy focal person | 35 | 9 | Male |
| 9 | KII 9 | Health Care Provider | Health facility In-charge | 31 | 7 | Male |
| 10 | KII 10 | Health Care Provider | Health facility In-charge | 30 | 8 | Male |
| 11 | KII 11 | Health Care Provider | Health facility In-charge | 34 | 7 | Male |
| 12 | KII 12 | Health Care Provider | Leprosy focal person | 30 | 6 | Male |

observation of satellite clinics and health facilities. Interview questions were based on Nepal's leprosy operational guidelines [10] aligned with WHO guidelines [11] to explore the implementation fidelity of leprosy care and support.

The tools were initially developed in English and later translated into Nepali. The tools were pre-tested with two healthcare providers and one SHG from similar settings to check clarity and cultural relevance, approximately one week before the actual data collection. Based on pre-testing, minor modifications were made to the flow of questions and wording to ensure clarity during data collection. The local facilitators assisted during an interview to ensure the participant's comprehension. Data were recorded using a digital audio recorder, and principal investigators took field notes. All audio recordings and transcriptions were manually transcribed verbatim by trained independent research assistants and cross-verified with field notes to ensure data quality. The KIIs lasted approximately 30 minutes, whereas FGDs were completed in 60–70 minutes. Participants were encouraged to elaborate on detailed experiences to provide rich information. The data was collected until the data saturation was obtained with the last KII as guided by [19].

### Data analysis

The data analysis followed Braun and Clarke's thematic analysis framework involving systematic steps to ensure the credible interpretation of the findings and their linkages to the research objectives [20]. First, the research team familiarised themselves with the data by reviewing transcripts and observational notes. The data were transcribed verbatim in Nepali and then translated into English. The English-translated versions were used for coding and thematic analysis. The research assistant and principal investigator validated the data after transcription and ensured its accuracy.

Initial codes were generated through line-by-line coding, capturing key elements such as diagnostic tools and healthcare provider skills. These codes were reviewed and grouped into broader subthemes, such as "Accessibility of the service," which summarised challenges like diagnostic kit availability, referral mechanisms, and human resource gaps. Subthemes were further aggregated to form broader themes and refined to ensure alignment with the data, to support the findings. For instance, participants highlighted delays caused by insufficient diagnostic kits and the financial burden of accessing tertiary care.

To support the analysis, data were coded using qualitative analysis software. We used Deedose software to organize and manage the coded data. Inductive coding allowed for the generation of new codes, as stated in Table 3 and a codebook was developed to ensure consistency among researchers. All types of data (KIIs, FGDs and observation) were analysed using the same thematic analysis with special attention paid to the nature of the data source. A manifest analysis was conducted to analyse the qualitative data and to interpret a comprehensive understanding of facilitators and barriers to the fidelity of leprosy care and support implementation. A preliminary analysis was shared with stakeholders for feedback and data triangulation. These findings were interpreted in the context of national leprosy operational guidelines [10].

### Researcher reflexivity and positionality

The lead investigator (SN) is a public health professional with over six years of work experience in neglected tropical diseases, including leprosy, at the current study site. SN has completed a Master's in Public Health (MPH) with a specialisation in Implementation Research (IR). Having witnessed firsthand the barriers to leprosy services, SN approached the research process with extensive knowledge of both policy and ground realities. The data collected in the study were generated in the form of codes and guided by Braun and Clarke's thematic analysis framework, which influenced the development of themes, including barriers to leprosy care and support and healthcare providers' adherence to leprosy services.

AP and RAA are a professor and a senior epidemiology and implementation research expert, respectively, with over a decade of expertise in qualitative research. Their specialisation in infectious disease control and health service research contributed significantly to ensuring scientific and methodological rigour. Similarly, AT holds a double degree in MSc. in Public Health and M.A. in Gender Studies; AS is currently pursuing a Master's in Public Health and PCJ completed a

**Table 3. Comparison of national leprosy operational guideline and actual implementation of leprosy services.**

| Attribute | What national leprosy operational guidelines recommend | What was done (observed/reported) | Fidelity |
|---|---|---|---|
| Accuracy of diagnosis | Skin smear services are not mandatory for the health post, but should be available in the district hospital for diagnostic confirmation. | Skin smear services were absent in all 9 out of 10 health posts, so they referred to Butwal for skin smear confirmation. Diagnosis in health posts was based only on clinical signs. | Sufficient |
| Case management | Adherence to diagnosis using the three cardinal signs and standardised case management; MDT availability. | Eight out of ten HFs adhered to the cardinal signs for case diagnosis. Two HFs referred all cases without a full assessment. MDT was available in all health facilities. | Sufficient |
| Timeliness of services | Suspected cases should be managed promptly at the point of care; prompt MDT initiation | Most HFs managed suspected cases promptly. Two HFs referred all cases without an onsite assessment. All HFs promptly initiated MDT after diagnosis. | Sufficient |
| Complication management | Simple complications to be managed at the local health post; complex ones should be referred appropriately. | Eight out of ten HFs referred ulcer and lepra reaction cases to the referral centre instead of managing locally. | Insufficient |
| Follow-up and monitoring case registration | Follow-up should occur every 28 days; VMT/ST should be performed at 3 months. Verify medication use; track progress | All HFs performed 28-day follow-up via home visit, phone call and patient contact through FCHVs. Nine out of ten HFs performed Voluntary Muscle Testing (VMT)/ Sensory Testing (ST) at 3 months. All HFs recorded cases in the leprosy register and reported in the demographic health information system2 (DHIS2) software; All HFs verified medication use during follow-up. | Sufficient |
| Self-care involvement | Healthcare providers should demonstrate and teach self-care techniques effectively. | All HFs taught self-care verbally. Only three HFs demonstrated techniques. | Insufficient |

Master's in Public Health (MPH). SN, AT, AS and PCJ are familiar with Nepali language and culture and have strong experience in leading qualitative studies on the Nepalese context on a range of public health issues.

All co-authors reviewed the data collection tool and supported the analysis and interpretation of the data. Furthermore, collectively the co-authors agreed on the finalisation of a tool, development of codebook, development of sub-themes and themes, preparation, revision and submission of the manuscript. Throughout the study, all authors approached the study with cultural sensitivity, recognising that their positionality could influence how data is interpreted and analysed.

### Ethics statement

The study received an ethical approval letter from the Ethical Board Committee, Universitas Gadjah Mada, Indonesia, (Reference no: KE/FK/0291), Nepal Health Research Council, (Reference no: 1078), and Health Office Rupandehi (Reference no: 208). Ethical approval covered all study procedures, including the consent form. Verbal consent was taken and documented by the research team in the form of audio recordings, and written consent was obtained from participants before the recordings were taken and signed by the interviewer. Participants were informed about the study's purpose, confidentiality, voluntary participation, and the right to withdraw at any time. Proxy participation was not permitted, and only participants who met the inclusion criteria and provided informed consent were involved in the research.

## Results

This section categorises themes and subthemes as per research objectives. It is further categorized into three major themes and 10 sub-themes.

### Theme 1: healthcare provider's adherence to leprosy services

**Case and complication management for disability prevention.** The primary health institution provides preventive services for complicated cases such as ulcers and lepra reactions by conducting assessments via VMT/ST, counselling,

and simple ulcer dressing while also teaching self-care skills. Following this initial care and support, people with leprosy are referred to tertiary hospitals for specialized care and management.

*We conduct assessments through VMT/ST and provide simple dressings for dead tissue removal and teach self-care skills to prevent complications. If complications occur, we refer complicated ulcers and lepra reactions to Anandaban Hospital for specialised treatment (KII 9, healthcare provider, 31-year-old, male).*

**Adherence to leprosy operational guidelines.** Almost all healthcare providers are aware of the national leprosy operational guideline/WHO guidelines for treating and diagnosing leprosy and consistently follow them. They also have adequate knowledge about the requirement to conduct VMT/ST every three months.

*I am familiar with leprosy operational guidelines. The duration of MDT depends on the leprosy categorization, it could range from 6 to 12–24 months, etc. We also do VMT/ST at follow-up visits every three months (KII 8, healthcare provider, 35-year-old, male).*

Similarly, the national leprosy operational guidelines are easily accessed on the internet and are designed to be user-friendly. This makes it easier for healthcare providers to follow protocols and gives them clarity.

*The current available national leprosy operational guidelines are easily accessible and provide clear instructions to the healthcare provider regarding the management of leprosy. Thus, it is easy to follow the guidelines and it is easily available on the internet (KII 8, leprosy focal person, 35-year-old, male).*

The diagnosis and treatment of leprosy are greatly hampered by the unavailability of skin smear tests due to the absence of qualified lab personnel and essential diagnostic kits in the health facilities. The need for people with leprosy to travel to referral institutions like Butwal places an enormous burden on the healthcare system and individuals with leprosy.

*Despite having a lab facility, skin smear tests are not available in the health facility. Therefore, to do skin smear tests, we must refer them to Butwal because a lab person in a health facility is not trained to perform the skin smear test (KII 3, healthcare provider, 48-year-old, female).*

**Follow-up and monitoring of case registration.** Once leprosy cases are confirmed, the healthcare provider records and registers the case in the leprosy treatment register in the local health post. The case-based details are reported in the DHIS2, the national data repository. Healthcare providers follow various strategies to ensure the treatment process in follow-up. These include contacting via phone calls and taking FCHVs support to help verify medicine consumption and provide additional support.

*Once there is leprosy confirmation, we enrol them in MDT and record details in the register. Initially, we provide medicine for 28 days and ask them to come with the provided medicine for follow-up. If they do not come for regular follow-up, we directly call them or ask an FCHV to bring them to the health post. During follow-up, we cross-verify consumed MDT and track progress (KII 1, healthcare provider, 33-year-old, male).*

### Theme 2: experience of healthcare providers to provide quality leprosy services

**Accessibility of the service.** Healthcare providers adhere to national leprosy operational/WHO guidelines during case diagnosis and treatment. They initially look for cardinal signs, including hypopigmented skin lesions with sensory deficits

and peripheral nerve involvement. When required, referrals are made to tertiary hospitals. When a confirmed diagnosis is made, people with leprosy are enrolled in MDT immediately.

*We look for the cardinal signs to diagnose leprosy, like hypopigmented skin lesions with a definite sensory deficit and involvement of peripheral nerves. If there is a suspect, we refer them to Butwal for skin smear confirmation. Based on the presence of skin lesions and nerve involvement, we categorize them as PB and MB cases and start MDT (KII 4, health facility in charge, 42-year-old male).*

Participants receive a wide range of services from healthcare facilities. Free MDT and basic dressing for minor ulcers are among these services. Healthcare providers also provide counselling on complications, techniques for self-care, and pain management of lepra reactions. For more complex situations, such as severe lepra reactions and ulcer care, they are referred to specialised centres.

*We offer a range of care and support services from the health post to prevent and manage disabilities. These services include free MDT, simple ulcer management, counselling to prevent complications, routine follow-up, contact tracing and referral services to complicated cases (KII 7, healthcare provider, 33-year-old, female).*

**Timeliness of services.**  Immediate action upon any leprosy suspects to the client visiting the outpatient department. They start the assessment procedure along with maintaining the patient's privacy, confirm the diagnosis, and start MDT along with counseling.

*When a client in the outpatient department is suspected of having leprosy, we initiate the assessment process immediately with due respect to privacy. This helps to avoid delays in services in health posts. After confirmation of the diagnosis, we start MDT immediately (KII 9, healthcare provider, 31-year-old, male).*

**Skills of health care providers.**  Healthcare providers needed training to manage leprosy complications. Having completed leprosy training, the majority of healthcare providers are better equipped to handle leprosy cases. However, they believe that, even with years of expertise, specialised training in complication management is still necessary to deliver high-quality care.

*I completed leprosy training over 15 years ago. I feel more confident in managing and dealing with leprosy cases now that I have a better understanding of leprosy. However, due to the advancements in leprosy care and support, I require specific training on complications management, particularly for managing lepra reactions and ulcer care (KII 4, health assistant, 42-year-old, male).*

A few healthcare providers emphasized onsite coaching as an effective method of boosting healthcare provider capacity. They are receiving direct coaching in the field to help them better manage leprosy complications and ensure quality care.

*Onsite coaching from the Health Office, Palika, and experts from external development partners play a key role in the care and support of people with leprosy for disability prevention and management (KII 5, health assistant, 29-year-old female).*

### Theme 3: participants' responsiveness to the leprosy services

**Satisfaction towards services among people with leprosy.**  Local health facilities lack the necessary resources to manage complex leprosy complications such as ulcers and reactions effectively. Due to the limited scope of services

available at nearby health posts, people with leprosy face difficulties in treating complicated cases such as ulcers and lepra reactions. Participants shared that due to their disabilities, they face difficulty in travelling and also experience financial burdens.

*At our local health post, complications such as lepra reactions and ulcers are not treated due to insufficient resources and medications to manage complex cases. This has caused difficulties for us to travel to specialised hospitals like Anandaban periodically for treatment (FGD 1, Self Help Group (SHG) member (N6), 25-year-old, female).*

People with leprosy find the requirement to take daily medication challenging and burdensome. They express frustration that if doses of medicine are missed, they have to restart the treatment from the beginning. They report experiencing side effects from leprosy medications. Some get annoyed with healthcare providers because, even after taking treatment for a long time, they can't see an improvement in regaining sensation in affected areas or the resolution of skin nodules.

*Some complain of experiencing side effects like extreme weakness and changing the appearance of their skin while a few complain that even after taking the medication for an extended period, they have not regained sensation in their affected areas and nodules present in their skin, especially their face haven't recovered fully like before, and they become angry (KII 4, health facility in-charge, 42-year-old, male).*

People with leprosy face severe stress and social challenges as an outcome of the side effects, especially changes in skin appearance, which leads to discrimination and stigmatization. The in charge of the health institution mentioned that one individual with leprosy even threw their medicines into the river due to side effects, such as changes in skin colour.

*During a routine follow-up, we inquired with the spouse of an individual with leprosy about the MDT consumption. We found that despite counselling at the beginning individual with leprosy had thrown away the medicine strip into the river due to the side effects. After counselling the individual with leprosy and her spouse, we restarted the MDT (KII 2, health facility in-charge, 34-year-old, female).*

**Level of engagement in self-care.**  Healthcare providers provide comprehensive counselling, covering subjects such as drug side effects, self-care techniques, emotional support, and complications. This service has significantly contributed to emotional support, relief and treatment adherence.

*After leprosy confirmation, I was so stressed but my healthcare provider counselled me properly. He said this disease is completely treatable and no need to be stressed and informed about possible leprosy complications. He advised me to take medicine daily and informed me about self-care practices such as protecting my hands and feet which will help me with disability prevention. So I felt a huge relief from his counselling (FGD 1, SHG member (N7), 28-year-old, female).*

**Perceived stigma in accessing leprosy services.**  People with leprosy frequently ask healthcare providers to maintain the privacy of their condition because of the stigma. Being hesitant to reveal their status makes it more difficult to get social support and carry out contact tracing for the healthcare provider. Fear of discrimination leads some individuals to avoid seeking medical care or to travel long distances to access services anonymously.

*Due to stigma, I avoid medical care in our local health facilities due to fear of being discriminated against by relatives, neighbours and community members, As a result, I have to travel a long distance to receive the same medical services. My fear stops me from receiving the timely care I need for my disease, which eventually results in deteriorating health (FGD 2, SHG member (H2), 55-year-old, male).*

Some people with leprosy have even lost their jobs due to the disclosure of their leprosy diagnosis. Now, they do not want to continue treatment because they fear losing their current job.

*We attempted to trace the social contact of one of the under-treatment individuals with leprosy at our health post. However, the individual requested not to trace in his current workplace, as he lost his job at his previous workplace after his leprosy status was revealed, and he does not want to risk his current job again (KII 11, healthcare provider, 34-year-old, male).*

**Barriers and facilitators of leprosy services.** The people with leprosy expressed that they face difficulty in travelling long distances during referrals due to leprosy-related complications such as ulcers and reactions.

*The major barrier to accessing leprosy services is transportation for mobility due to leprosy-related complications such as ulcers and reactions. These complications are not managed in the local health post and it is difficult for us to travel to Kathmandu for complication treatment (FGD 1, SHG member (N2), 50-year-old, male.)*

People with leprosy are discouraged from obtaining treatment because of the expensive transportation cost to the tertiary hospital in Kathmandu, which worsens their health condition. Additionally, people with leprosy frequently request financial support to travel to specialized centres like Anandaban Hospital in Kathmandu.

*There is no additional financial support from a municipality to the people with leprosy, as they expect financial support from us while being referred to the referral centre in Kathmandu. It is extremely difficult to counsel them for the referral to Kathmandu for tertiary care (KII 1, healthcare provider, 33-year-old, male).*

The people with leprosy refused to get medical treatment because of his superstitious belief that his condition was due to a divine curse, which could only be healed through worship. Such beliefs lead to delays in seeking appropriate medical care, thereby hindering effective treatment and management of the diseases.

*One patient diagnosed with leprosy refused to start MDT even after counselling. He believed that his condition was due to a curse from God, and he needed to perform some pooja to heal it. He ran away to live with a relative in India. It took us a full year to convince him to start multi-drug therapy (MDT) and get the treatment he needed (KII 3, leprosy focal person, 48-year-old, female).*

The lack of disability-friendly infrastructure severely hinders people with leprosy's access to healthcare facilities. In addition, the lack of separate examination rooms makes it difficult to maintain the privacy of people with leprosy, which makes them uncomfortable and reluctant to go to health facilities for leprosy diagnosis and follow-up care.

*We don't have adequate rooms in health posts, so it is difficult for us to provide leprosy services in limited spaces, as people with leprosy need to be handled sensitively. Maintaining privacy for the diagnosis in outpatient department is difficult for us, and we require a separate room for the diagnosis and complication management of leprosy (KII 5, leprosy focal person, 29-year-old, female).*

One of the informants highlighted that health facilities encounter challenges because the local government supplies MDT solely based on the number of registered leprosy cases, without accounting for emergencies.

*The swift supply of MDT from the Palika is a challenge as MDT is provided to us on a case-by-case basis, which means that during times of high caseload in health post, we have to borrow MDT from neighbouring health facilities. This makes it difficult to maintain a consistent supply chain of MDT every month (KII 12, leprosy focal person, 30-year-old, male).*

The presence of experienced, trained, and motivated staff significantly enhances the provision of care and support for people with leprosy. Leprosy training provided by partner organizations in coordination with the Nepal government has boosted the confidence and skills of healthcare providers, enabling them to manage complications and prevent disabilities effectively.

*The provision of leprosy training to healthcare providers by partner organizations in coordination with Health Office Rupandehi has made it easier to provide leprosy care and support as healthcare providers have been confident after receiving training and motivated to support people with leprosy for the disability prevention and management (KII 8, healthcare provider, 35-year-old, male).*

Involving FCHVs during follow-up visits has proven beneficial in ensuring adherence to medication regimens. Similarly, FCHVs assist in identifying hidden cases and following up on cases, including conducting home visits if necessary. They play a key role in detecting hidden leprosy cases within the community and ensuring that people with leprosy adhere to their treatment regimens.

*FCHVs have been supporting us in finding new leprosy cases as they refer the suspected leprosy cases from the community to our health facility. In this way, they play a key role in searching for hidden leprosy cases in the community (KII 1, leprosy focal person, 33-year-old, male).*

Health institutions provide services and counselling in the native language, which facilitates effective communication and ensures there are no language barriers in accessing leprosy services. In this way, people with leprosy do not feel language barriers.

*We haven't encountered any language barriers from the healthcare providers as they are familiar with the Abadhi language. They communicate effectively about the available leprosy services, and their proper counselling has made it easier for us to understand and access leprosy services for disability prevention and management (FGD 2, SHG member (H6), 25-year-old, male).*

## Discussion

This is the first study to qualitatively explore implementation fidelity in leprosy care and support for disability prevention and management in the Rupandehi district of Nepal. The study's findings highlight strengths such as sufficient adherence to national leprosy operational guidelines [10], ensuring the prompt initiation of MDT, case diagnosis, complicated cases referral, and regular follow-up. However, poor adherence was observed in managing lepra reactions, ulcer cases, and self-care activities.

The findings of this study highlight the importance of Nepal's holistic approach to leprosy care and support, particularly through free MDT provision, free treatment, and financial incentives (e.g., a cash-based incentive of 1000 NPR upon completing the MDT treatment), which have improved treatment adherence and reduced stigma. These findings align with a study from Brazil, where a cash transfer programme improved treatment adherence and cure in multibacillary cases [21].

To reach the endemic areas and marginalized populations, the participants of this study have identified the critical role of satellite clinics in screening, diagnosis and managing complications. This important initiative to facilitate leprosy has successfully reduced the financial burden associated with travelling to the tertiary hospital in Kathmandu for routine follow-up for disability prevention and management by providing quality leprosy care provision in the local communities.

Referrals to monthly satellite clinics and tertiary hospitals, such as Anandaban Hospital in Kathmandu, ensured specialized care for complex leprosy cases. The use of referral slips from health facilities with accurate diagnoses and

contact details made the referral process simpler to receive specialized care and support. Furthermore, the engagement of FCHVs in the treatment process was found to have assisted in treatment adherence by suspecting and referring the cases, and following up on leprosy cases. These approaches enhance the quality and accessibility of leprosy services.

Healthcare providers engaged in complication management through quarterly VMT/ST, basic pain management, self-care training and timely referral of complicated cases to Anandaban Hospital. This shows a coordinated approach to disability prevention aligning with several similar studies [22,23]. However, local health facilities' lack of resources and expertise for complication management of lepra reaction and ulcer care remains a major challenge. This inadequacy resonates with findings from other low-resource settings where healthcare providers often face similar constraints [24].

The majority of healthcare providers showed an extensive knowledge of cardinal signs and MDT regimens which demonstrate sufficient adherence to national leprosy operational guidelines [10] aligning with WHO guidelines [11]. This is foundational knowledge for healthcare providers for an accurate diagnosis. On the contrary, findings from relevant studies [25–28] revealed that medical professionals involved in the treatment lacked case management and treatment guidelines awareness. Our findings suggest that readily available and user-friendly treatment guidelines facilitate the primary reason for the healthcare provider's adherence. The healthcare provider maintained standardized care across several healthcare facilities due to national leprosy operational guidelines clarity. In addition, adherence to MDT regimens for specified durations (6, or 12 months) ensured appropriate and effective treatment. In certain cases, particularly those with persistent complications treatment durations were extended based on clinical judgement. These treatment extensions were implemented following national leprosy operational/WHO guidelines, and healthcare providers' adherence to treatment extensions helped to improve patient outcomes.

The availability of free MDT at public facilities is crucial in ensuring treatment adherence among developing countries with limited access to health services [14,28]. Similarly, our study findings resonated with the availability of free MDT and their immediate enrolment for improving treatment adherence. Furthermore, follow-up strategies such as regular contact through phone calls, home visits, and involvement of FCHVs and treatment supporters ensured continuous care, compliance and treatment adherence.

The implementation strategy practised in the Rupandehi such as the capacity building of healthcare providers prioritizing the leprosy pocket areas, regular follow-up visits including VMT/ST, pill count, cross verification of MDT strips and involvement of family members as treatment supporters could be replicated in other resource-constrained settings globally to improve healthcare provider adherence to leprosy services. Furthermore, practical approaches like medicine strips requirement during follow-up, pill counts for cross verification and monitoring the treatment progress closely resonate with strategies proposed in a similar study [29].

Skin smear examination is crucial for accurate diagnosis, classification, monitoring of treatment, and disease severity assessment [30,31]. However, our study findings revealed several gaps in accurate diagnosis at the local health post despite having basic lab facilities. The absence of skin smears due to the unavailability of trained lab personnel and diagnostic kits at local health posts has forced people to travel to hospitals in Butwal for skin smear tests leading to financial and logistic burdens on both the people with leprosy and the healthcare system. So, the healthcare providers in this study have expressed their disappointment over the systematic barrier and stressed the dire need for skin smear training for lab personnel of health posts and diagnostic tools to perform skin smear tests locally.

Few studies [27,32–34] have demonstrated the effectiveness of leprosy training to increase existing knowledge. Healthcare providers appreciated the benefits of training programmes as training equipped them with the necessary abilities to manage leprosy. However, as leprosy care advances, there is an urgent need for complication management training, focusing on treating lepra reactions and ulcer care. This reflects a regional need across South Asia in which misdiagnosis and several consultations before diagnosis were major contributors to the delay [35].In contrast, our findings demonstrate healthcare providers handled complications by doing assessments, providing basic ulcer dressing, and imparting self-care skills.

The majority of SHG respondents were satisfied with the timely treatments, comprehensive assessments, counselling, contact tracing and the consistent availability of MDT in health facilities. This finding is consistent with a few studies [4,36] that highlighted timely and comprehensive care in managing leprosy. However, our observation and findings suggest that people with leprosy faced difficulties in treating complications such as ulcers and lepra reactions. The primary reason is due to mobility issues, and also financial burdens. On the other hand, a significant number of participants reported experiencing severe adverse effects, including acute weakness, skin discoloration, nausea, and dizziness, which the healthcare providers inadequately managed. These findings align with a few studies [7,29,36–38] that highlight the need for better management of adverse drug reactions to improve treatment adherence.

In Nepal, MDT duration varies based on case type: 6 months for paucibacillary cases and 12 months for multibacillary cases. However, complications like lepra reactions or ulcers require treatment extensions. Our research findings show that participants expressed frustration with the daily pill burden, significantly when treatment was prolonged, which led to nonadherence. For example, the healthcare provider narratives exemplified the incident of an individual who 'threw away' MDT because of the side effects and pill burden. Furthermore, our study findings aligned with studies from India and Brazil in which patients were found to be non-adherent and admitted to forgetting to take their medicines, being careless at times about taking their medicines and patients admitted to stopping the medicines [38,39]. Global studies indicate that treatment extensions can strain patient compliance if not supported by strong assistance measures [37].

Our findings suggest a significant pill burden as a result of the drug regimen may impact adherence and overall quality of life of people with leprosy. Healthcare providers emphasized the necessity of comprehensive counselling and family involvement to mitigate these challenges, aligning with findings from [14], where patient adherence was closely linked to provider fidelity to follow-up protocols. Numerous studies carried out by [13,15,40–42] are consistent with our study findings regarding the pill burden and patient non-adherence to the medicine regimen.

The findings of our study indicate that demographic characteristics, particularly age and remote residence, play a critical role in access to leprosy diagnosis and continuity of treatment. Older adults and people from remote areas experienced delays in leprosy services due to transportation difficulties and a lack of awareness. Similar urban-rural disparities have been confirmed in India and Bangladesh [43,44].

Stigma and discrimination were identified as the key attitudinal barriers that compromise treatment adherence and influence people with leprosy's willingness to reveal their condition and seek timely care and support. Due to fear of discrimination and social exclusion, many people with leprosy in this study chose to seek anonymous care and support from distant facilities rather than going to nearby health facilities while few people refused treatment due to superstitious beliefs hindering the effectiveness of treatment. These kinds of superstitious beliefs often delay leprosy care and ultimately result in disability. This finding is consistent with a study [45] in which superstitions had a strong impact on health-seeking behaviour in people with leprosy.

Apart from this, high out-of-pocket expenditure, superstitious beliefs and difficulty in travelling, and institutional barriers (such as inadequate resources, lack of skilled human resources and inconsistent drug supply) hindered service delivery. Participants are discouraged from getting essential care due to the high cost of transportation and other associated costs. Our study findings are consistent with these studies [46–48]. The institutional barriers, such as insufficient infrastructure and resources also hindered the leprosy services. Furthermore, privacy concerns in the absence of separate examination rooms also discouraged from seeking treatment. These findings are consistent with a study done by [49]. To address this issue, healthcare providers must be sensitised to enhance their communication skills and responsiveness.

Policy-level improvements are needed to improve the fidelity of leprosy care delivery in health post of Rupandehi district. Prioritized interventions, such as investing in skin smear services at the local health post, are critical to ensuring swift and accurate diagnosis. Integrating complication management into leprosy training is critical for equipping healthcare providers with complication management skills. Furthermore, strengthening physical infrastructure at health posts allows for patient confidentiality and comfort, which are crucial for reducing stigma as well as increasing treatment adherence.

Our study findings identified several facilitators of leprosy services such as adequate human resources, involvement of treatment supporters, effective communications, and the role of external development partners. The involvement of family

members in follow-up visits to monitor progress and treatment adherence is another key facilitator to follow MDT regimen, motivation and emotional support for well-being and self-care. Our study findings aligned with study [50], which highlighted the role of psychosocial support in leprosy recovery.

While comparing the health facilities, we found fidelity of implementation varied significantly across the health facilities, highlighting the key gaps in service provision that impact the leprosy services. The majority of health facilities show sufficient fidelity, the main reason is adherence to national leprosy operational/WHO guidelines which led to the delivery of standardized leprosy services. On the other hand, few health facilities show poor fidelity since they follow national leprosy operational/WHO guidelines inadequately and rely heavily on referral services for leprosy management. The inconsistency in service delivery highlights the gaps in training, resource allocation and treatment protocol adherence.

## Strengths and limitations of the study

This study only covered the participants from one of the leprosy endemic districts, i.e., Rupandehi district which may not be generalised in some cases, however, the findings provide important policy input for the policymakers. Furthermore, this study focused on healthcare providers from local health facilities who provided the same leprosy care and support services, limiting insights from another setting, such as a hospital, which may offer different resources and strengths in leprosy care and support services.

The researcher initially intended to select an identical number of healthcare providers, i.e., leprosy focal person and health facility in charge of interviews until data saturation. However, challenges such as staff transfer and the new appointment of a leprosy focal person resulted in 12 healthcare providers from 10 health facilities being interviewed.

## Conclusion

The role of health care providers is crucial not just for early diagnosis, and complications management but also to provide holistic support beyond existing guidelines, including promoting self-care and addressing the barriers. Healthcare providers have been found to adhere sufficiently to Nepal's leprosy operational guidelines. Accurate and prompt diagnosis, robust follow-up, and referral centre accessibility are major strengths, but challenges such as unskilled lab personnel in local health facilities, and over-dependence on referral centres remain major concerns. Participants value healthcare providers' timely care, accurate assessment, counseling and self-care engagement. However, they have major obstacles due to the daily pill burden of MDT, drug side effects and leprosy complications. Furthermore, the leprosy stigma complicates the accessibility and treatment adherence.

The major barriers to leprosy services include referral financial hardship, complication management, drug side effects, and institutional obstacles. However, facilitators include treatment supporter involvement, effective communications, the role of an external development partner, and support from the local government to enhance service accessibility. Addressing these barriers while leveraging facilitators is crucial to achieving the global target of Zero disability set by the global leprosy strategy 2021–2030.

## Supporting information

**S1 Table. Codebook used for qualitative data.**
(DOCX)

**S2 File. Informed consent form.**
(DOCX)

**S3 File. Research tools.**
(DOCX)

## Acknowledgments

We thank the WHO/TDR postgraduate scholarship program/ Universitas Gadjah Mada, Indonesia for providing scholarship. This study would like to acknowledge the study participants, The Leprosy Mission Nepal, and Health Office Rupandehi for their support during the study process. We express our sincere gratitude to Sunil Nepal, Ghanshyam Pandey, Anita Mahotra, Bishnu Dhungana and Sita Ale for their invaluable support during the research process.

## Author contributions

**Conceptualization:** Sudip Nepal, Ari Probandari, Riris Ahmad Andono.

**Data curation:** Sudip Nepal, Ari Probandari, Riris Ahmad Andono.

**Formal analysis:** Sudip Nepal, Ari Probandari, Anisha Shrestha, Riris Ahmad Andono.

**Methodology:** Sudip Nepal, Ari Probandari, Riris Ahmad Andono.

**Supervision:** Ari Probandari, Riris Ahmad Andono.

**Validation:** Sudip Nepal, Ari Probandari, Amit Timilsina, Riris Ahmad Andono.

**Writing – original draft:** Sudip Nepal, Amit Timilsina.

**Writing – review & editing:** Sudip Nepal, Ari Probandari, Amit Timilsina, Anisha Shrestha, Prakash Chandra Joshi, Riris Ahmad Andono.

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
