## [Decision Letter · Decision Letter 0]

Dear Dr. Nepal,

We look forward to receiving your revised manuscript.

Kind regards,

Veincent Christian Pepito

Academic Editor

PLOS ONE

Journal Requirements:

2**.** Please include a complete copy of PLOS’ questionnaire on inclusivity in global research in your revised manuscript. Our policy for research in this area aims to improve transparency in the reporting of research performed outside of researchers’ own country or community. The policy applies to researchers who have travelled to a different country to conduct research, research with Indigenous populations or their lands, and research on cultural artefacts. The questionnaire can also be requested at the journal’s discretion for any other submissions, even if these conditions are not met.  Please find more information on the policy and a link to download a blank copy of the questionnaire here: https://journals.plos.org/plosone/s/best-practices-in-research-reporting. Please upload a completed version of your questionnaire as Supporting Information when you resubmit your manuscript.

3. In the ethics statement in the Methods, you have specified that verbal consent was obtained. Please provide additional details regarding how this consent was documented and witnessed, and state whether this was approved by the IRB

6. Please include captions for your Supporting Information files at the end of your manuscript, and update any in-text citations to match accordingly. Please see our Supporting Information guidelines for more information: http://journals.plos.org/plosone/s/supporting-information .

**Additional Editor Comments:**

Dear authors,

Thank you very much for taking on the burden of doing implementation research for leprosy in Nepal. The paper is good in theory but needs to be revised so it is more faithful to the title. Please see my comments below:

1. The paper talks about implementation fidelity but does not bother comparing and contrasting with country clinical practice guidelines or leprosy control program operations manual or WHO best practices for leprosy. How are you able to conclude that there was implementation fidelity (or there was none)? Please consider adapting the methodology used here where we compared what the facilities found with what is the recommended practice by our ministry of health and the World Health Organization: https://actamedicaphilippina.upm.edu.ph/index.php/acta/article/view/409

2. What is the theoretical framework used in analyzing the study? How were the themes generated and what is the basis for presenting these themes?

3. What is the reflexivity and positionality of the authors as they did this study? Please add this to the manuscript.

4. While you are discussing adherence to protocols, can you also discuss treatment extension, fidellity of healthcare providers in keeping with treatment guidelines, and its effect on patient adherence to medication?

https://www.cell.com/heliyon/fulltext/S2405-8440(21)01382-7

5. Discussing self-care for leprosy is good but I think there is a need to be cautious about what self-care practices are good and what self-care practices are bad:

https://academic.oup.com/heapol/article/38/2/205/6798855?login=false

https://www.emerald.com/insight/content/doi/10.1108/ijhg-01-2023-0008/full/html

6. While it was done in passing, can you please give a more detailed description of Nepal's healthcare system and the provision of leprosy services to familiarize the international reader about the paper? Specifically, please mention how long is the travel time from you to referral hospitals and other context to fully appreciate the issues that you have discussed, or if you are from an urban area or a rural area?

Reviewers' comments:

Reviewer's Responses to Questions

**Comments to the Author**

1. Is the manuscript technically sound, and do the data support the conclusions?

Reviewer #1: Partly

Reviewer #2: Yes

2. Has the statistical analysis been performed appropriately and rigorously?

Reviewer #1: Yes

Reviewer #2: N/A

3. Have the authors made all data underlying the findings in their manuscript fully available?

Reviewer #1: Yes

Reviewer #2: Yes

4. Is the manuscript presented in an intelligible fashion and written in standard English?

Reviewer #1: Yes

Reviewer #2: Yes

Reviewer #1: Implementation fidelity in leprosy care and support for disability prevention and

management in Rupandehi, Nepal: A qualitative study

<abstract>

Background

Your background section emphasizes the importance of implementation fidelity but lacks a clear rationale for studying it in the context of leprosy care in Nepal. To strengthen the justification, briefly explain why exploring implementation fidelity is crucial for improving leprosy care in Nepal. Also, a shorthand definition to implantation fidelity, could be beneficial to the readers unfamiliar to the topic.

Methodology

Your methodology section effectively outlines a numerous qualitative methods used, but specifying the number of participants would enhance clarity. Also, if applicable, please rearrange your sentences in the sequences the event takes place. For example, if sampling design is conducted prior to data collection. Also, mention the timeline the study was conducted.

Although ‘thematic data analysis’ is a used frequently, I think the correct term for qualitative research method is ‘thematic analysis.’

Results & Conclusions

As per author guidelines, you are discouraged to use abbreviation in the abstract. Please revise accordingly.

After reading your results and conclusions section in the abstract, I would like to suggest that you should consider revising the title of your study. Is your study specific to the role of healthcare providers in leprosy prevention and management or is it about overall implementation fidelity in leprosy care?

Please ensure the consistency of ‘tense’ in your sentences.

Overall

Please refer to the PLOS ONE author guidelines. Your abstract is at least 23 words above the permitted word limit. Your abstract is comparatively well-written in the introduction and methods section, while you may consider tightening your sentences in the results and conclusions sections, particularly focusing to reduce redundancy. Otherwise, this is a good abstract.

Introduction

First paragraph

Please cite a source that listed 22 prioritized countries on leprosy. As Nepal does not locate under Southeast Asia, replacing the regional prevalence of Southeast Asia to South Asia would be helpful. If this dataset is not available, you may cite the prevalence in neighboring and some South Asian countries. The sociocultural context of Southeast Asia may not closely resemble that in South Asia. Also, please quantify to what extent the cases of leprosy was reduced during the COVID-19 pandemic. Then, what’s the post-pandemic status of leprosy reporting?

Second paragraph

Please begin with presenting leprosy as a significant health problem in the history of Nepal, and then what the current status is. While you present the facts and figures, please make it comprehensible for the readers. For example, first clarify if leprosy is a disability? Disability and disease are different constructs. Then, explain the meaning of MDT. How severe are cases of leprosy of Grade 2 Disability and how is it different than other Grades?

It is evident that since 2010 leprosy was not a significant health problem at national level. However, how was/is the burden of leprosy in Rupandehi, Nepal? If this is not at a significant level, what is the significance of this study? Or, was/is the prevalence of leprosy in the study area underestimated? If yes, for what factors?

Third paragraph

Explain what health post in Nepal is. Is it the lowest level of healthcare access to Nepali populations?

<suggestion> In the second paragraph, it is better for readers to explicitly specify how many times the data was collected or what months the data was collected. The current version needs to be clarified regarding when the enrolment started.

Overall

The introduction section has important information but lacks proper flow and some important background information. For example, it lacks regional nation vs region-specific information on leprosy. Adding a context of history of leprosy in Nepal and connecting it to the current status of leprosy can be helpful. Also, socio-cultural environment of Nepal has interplayed an important role on leprosy care and management, which was lacking. To streamline the flow, I suggest you consider revising your introduction as below:

- Present leprosy as a significant public health problem in Nepal’s history, and how why studying leprosy care and management is still important in the current context in spite of its lower prevalence?

- Discuss leprosy care and management from the light of socio-cultural context in Nepal and specific to your study area.

- Discuss how leprosy care is available and what are the major barriers to its prevention and treatment.

- Provide a strong rationale why your study is important given the current scenario of leprosy in the study area. How does your study in a local context be useful to the national level and what are its relevance to the global society? Mentioning that leprosy care and support implementation fidelity is discussed only in grey literature does not provide a strong rationale for your study. Contextualize it in a more robust way.

- One major limitation in the introduction was the unclear delivery of what the term ‘implementation fidelity’ is. If you think that your study is more about ‘implementation fidelity,’ you will need to clarify this jargon to the readers and centralize your introduction under this term.

Methods

Ethics statement

Please use proper punctuation. For example, reference no should follow with ‘:’ rather than ‘;’. You don’t need to abbreviate any term you have not used more than three times in your manuscript. Also, explicitly specify which health office in Rupandehi did you seek consent from? Is it the provincial health office, or any ministry or something else? Did you provide a copy of informed consent to the participants? And, did you receive consent before recording the interview? Were there any renumeration or benefits provided to the participants in exchange for their time and contribution to your research? Was proxy permitted? As you have integrated several data collection methods, were the approaches to ethical considerations similar or different to each specific method?

Study setting

Provide a citation to justify your statement that Rupandehi is one the of the leprosy-endemic districts of Nepal. When you specify the borders, be explicit to differentiate the national versus international borders. It might also be insightful to consider inserting map depicting the study area. While you have specified 1.2/10,000 prevalence, there are two important pieces of information missing: a) to be considered a rate, you need a time variable; prevalence and prevalence rates are two different ideas in epidemiology; and b) which year is this prevalence of?

Also the following sentences – The government of Nepal (GON) …………milestones of NLEP is not relevant under this subheading.

Study design and participant selection

I believe you do not require this subheading.

Study design

Correct the case from title to sentence case and be consistent throughout your manuscript.

Please simplify when you explain your study design – qualitative implementation research methodology using a multi-source, multimethod case study design – is very vague. Rather specify the exact approach of qualitative research you implemented, such as case study design, grounded theory, narrative study or something else. Please elaborate the method you used for data collection.

Participant selection

No comments.

Table 1

Please add a table note specifying the standard used for leprosy categorization. Interestingly, 100% of cases are multi-bacillary. Are all cases of leprosy Multi-Bacillary in Nepal or in the study area? If not, why did you lack variability in your study?

Study tool and data collection

Please detail out if your tools were designed in English or Nepali language? How did you confirm the validity of the translation? You have mentioned that the tools were pre-tested but did not detail out how and when? Please add these details and describe what changes were made following the pre-test? Did you utilize any software for transcribing the script?

Data analysis

Please cite the software. Also, explain whether you used Nepali or English (if translated) version for the data analysis. Specify what specific methods were used for focused group vs. KII vs. observation. If all of them were analyzed using same approach, explicitly mention it.

Results

Adherence to leprosy protocols and guidelines

Interestingly, no detail of what standard protocol (national/international/WHO) used to treat leprosy was provided. Any thoughts?

Also, it was mentioned that the skin smear test was not possible due to the unavailability of a lab technician. Elaborating this in discussion could be insightful. Does this mean that these local health units in Nepal have necessary pre-requisites, but only lack the specific experts?

Follow-up and monitoring of case registration

Please integrate some points on how leprosy registration in Nepal is performed and whether or not there is a practice of universal or centralized record for leprosy in Nepal.

The only significant comment I have for results section is it’s a bit lengthy. Try to shorten it.

Discussion

Good start, particularly the first paragraph.

My suggestion is that the cases during COVID-19 are different to those in normal period. So, exclude discussing COVID-19 as your study was conducted in 2024, unless you specifically want to discuss a section on leprosy care management during the pandemic. Similarly, avoid having references relevant to the COVID-19 period accordingly as the scenario would be very different in COVID-19 period.

Discussions are well-detailed out. However, it would be more relevant to discuss your points referencing literature relevant to countries, preferably South Asia,, with similar healthcare and sociocultural systems. Integrate some major constructs, such as age, urban/rural differentials may also play important factor in diagnosis and treatment to leprosy. Also, integrate some important policy measure improvements.

References

Please refer to the author guidelines to format your references in an appropriate way.

Please be very specific when you have headings, subheadings in your study. Follow author guidelines to correctly represent the cases and levels of headings/sub-headings.

Overall

Leprosy has traditionally been an important public health concern in Nepal, particularly in rural context. This study needs a revised title; otherwise, the study represents an important work, reflecting the gaps in diagnosing and treating leprosy in Nepal. Methodology is appropriate with different methods to cross-validate the information obtained. The introduction can be framed better with suggested improvements, while results and discussion sections seem too lengthy and need tightening.</suggestion></abstract>

Reviewer #2: Very well written paper. No obvious comments to add. Just to make clarification in the manuscript- the basic procedures like VMT/ST and skin smear tests are not performed in government health facilities in Nepal. This makes diagnosis of leprosy delayed. This can be accepted for publication.

**Do you want your identity to be public for this peer review?** For information about this choice, including consent withdrawal, please see our Privacy Policy

Reviewer #1: No

Reviewer #2: No

---

## [Author Response · Author response to Decision Letter 1]

27 Apr 2025

Editor’s Comment

1.Please ensure that your manuscript meets PLOS ONE's style requirements, including those for file naming.

Response:

Thank you. We have ensured that our manuscript meets PLOS ONE's style requirements. We have addressed in the main document and all those supporting files accordingly.

Response:A complete copy of the PLOS questionnaire on inclusivity in global research has been attached as supporting information.

3. In the ethics statement in the Methods, you have specified that verbal consent was obtained. Please provide additional details regarding how this consent was documented and witnessed, and state whether this was approved by the IRB

Response: The study received an ethical approval letter from the Ethical Board Committee, Universitas Gadjah Mada, (UGM) Indonesia, (Reference no: KE/FK/0291), Nepal Health Research Council (NHRC),(Reference no: 1078), and Health Office Rupandehi (Reference no: 208). Ethical approval covered all study procedures, including the consent process. Verbal and written informed consent was obtained from all participants. The consent form was approved by both ethical committees from Nepal and Indonesia, and participants were informed about the study's purpose, confidentiality, voluntary participation, and the right to withdraw at any time. Verbal consent was documented by the research team in the form of audio recordings and written consent was obtained from participants before the recordings were taken and signed by the interviewer, as per the approved ethical protocol.

# Page 17 and 18

From lines 292 to 299.

4. We note that the grant information you provided in the ‘Funding Information’ and ‘Financial Disclosure’ sections do not match.When you resubmit, please ensure that you provide the correct grant numbers for the awards you received for your study in the ‘Funding Information’ section.

Response: The primary author, Sudip Nepal, received a scholarship from the World Health Organization Tropical Disease Research (WHO-TDR) Special Postgraduate Programme in Implementation Research at Universitas Gadjah Mada (UGM), Indonesia, to pursue a Master's in Public Health(MPH), and this study was carried out as part of his thesis. So there is no specific grant number available for this thesis study. This is revised and updated in the funding information section.

Response:The data from this study are fully available within the paper and its supporting information file for open access during the initial submission. The data availability statement is corrected accordingly in the submission form.

Response:The necessary changes have been made, and the supporting information files have been renamed as per the journal's guidelines.

Additional editors comment

1. The paper talks about implementation fidelity but does not bother comparing and contrasting with country clinical practice guidelines or leprosy control program operations manual or WHO best practices for leprosy. How are you able to conclude that there was implementation fidelity (or there was none)? Please consider adapting the methodology used here where we compared what the facilities found with what is the recommended practice by our ministry of health and the World Health Organization: https://actamedicaphilippina.upm.edu.ph/index.php/acta/article/view/409

Response: Thank you for your insightful feedback. We recognize the importance of systematically comparing observed practices with both national and international guidelines. So, we have systematically compared our findings with standard National Leprosy Operational Guidelines (2075). Nepal's leprosy operational guidelines have been adopted and are aligned with WHO guidelines. The comparison has been presented in Supplementary Tables (S1 and S2), where we analysed healthcare provider adherence across key dimensions such as case management, adherence to leprosy protocol, follow-up, and self-care involvement. Any observed deviations from the recommended practices have been discussed and how the observed fidelity aligns or deviates from the national leprosy operational guidelines ensures the implementation fidelity.

2. What is the theoretical framework used in analyzing the study? How were the themes generated and what is the basis for presenting these themes?

Response:

Thank you for your question. The study utilized the Carroll et al. (2007) framework to explore the implementation fidelity of leprosy care and support for disability prevention and management. The framework consists of several domains, including: Adherence, dose, quality of delivery, participant responsiveness, strategies to facilitate and program differentiation.

In our study, we have explored the quality of delivery, participant responsiveness, strategies to facilitate and adherence.

When we generated the themes, we adapted the domains in the Carroll et al.' framework and confirmed the study objectives and themes were generated based on study objectives. For example: under the theme of Healthcare provider’s experience to quality leprosy services, there are sub-themes of explored accuracy, timeliness, and accessibility of services (linked to quality of delivery) as per the Carroll et al framework of fidelity.

Under the theme of Participant Responsiveness, there are sub-themes of self-care practices, Satisfaction towards services and stigma, Barriers and Facilitators: Identified institutional, financial, and social factors. (linked to participant responsiveness) as per the Carroll et al framework of fidelity.

Healthcare provider adherence to leprosy services: Assessed protocol compliance, case and complication management (linked to adherence) as per the Carroll et al framework of fidelity.

3. What is the reflexivity and positionality of the authors as they did this study? Please add this to the manuscript.

Response: “The researcher consistently reflected throughout the research process, critically examining their biases and assumptions. By maintaining awareness of these influences, they aimed to minimize bias and enhance the credibility of their findings. Furthermore, the researchers approached the study with cultural sensitivity, recognizing that their positionality could influence how data is interpreted and analysed. Therefore, they strived for objectivity while acknowledging how their backgrounds might shape their interpretations. Multiple data sources and interviewing diverse participants ensured the credibility and dependability of the findings”

This information is updated in the methodology section of the manuscript accordingly. #page 17

From lines 280 to 287

4. While you are discussing adherence to protocols, can you also discuss treatment extension, the fidelity of healthcare providers in keeping with treatment guidelines, and its effect on patient adherence to medication?

https://www.cell.com/heliyon/fulltext/S2405-8440(21)01382-7

Response: Thank you so much for the constructive feedback. We have now expanded the discussion to address the topics of treatment extension, healthcare providers’ fidelity to national treatment guidelines, and their collective influence on patient adherence to medication in manuscript.

“The use referral slips from health facilities with accurate diagnoses and contact details made the referral process simpler to receive specialized care and support. Furthermore, the engagement of FCHVs in the treatment process was found to have assisted in treatment adherence by suspecting and referring the cases, and following up on leprosy cases.”

“Healthcare providers engaged in complication management through quarterly VMT/ST, basic pain management, self-care training and timely referral of complicated cases to Anandaban Hospital.”

“The majority of healthcare providers showed extensive knowledge of cardinal signs and MDT regimens which demonstrate sufficient adherence to national leprosy operational guidelines. This is foundational knowledge for healthcare providers for an accurate diagnosis”

“Our findings suggest healthcare providers maintained standardized care across several healthcare facilities due to national leprosy operational guidelines clarity. In addition, adherence to MDT regimens for specified durations (6, or 12 months) ensured appropriate and effective treatment. In certain cases, particularly those with persistent complications treatment durations were extended based on clinical judgement. These treatment extensions were implemented following national leprosy operational/WHO guidelines, and healthcare providers' adherence to treatment extensions helped to improve patient outcomes.”

“The follow-up strategies such as regular contact through phone calls, home visits, and involvement of FCHVs and treatment supporters ensured continuous care, compliance and treatment adherence”

# page no 33 to 35 From lines 565 to 600.

5. Discussing self-care for leprosy is good, but I think there is a need to be cautious about what self-care practices are good and what self-care practices are bad:

https://academic.oup.com/heapol/article/38/2/205/6798855?login=false

https://www.emerald.com/insight/content/doi/10.1108/ijhg-01-2023-0008/full/html

Response: Thank you for constructive feedback. Good self-care and bad self-care have been incorporated in the introduction paragraph.

“ Individuals may adopt harmful practices such as using inappropriate oils or herbal remedies, excessive soaking of wounds, or delayed wound care. Good self-care practices like routine wound cleaning, protective footwear, and regular self-inspection can all significantly lower the risk of ulceration and disability.

#Page 7

From lines 141 to 146

6. While it was done in passing, can you please give a more detailed description of Nepal's healthcare system and the provision of leprosy services to familiarize the international reader about the paper? Specifically, please mention how long is the travel time from you to referral hospitals and other context to fully appreciate the issues that you have discussed, or if you are from an urban area or a rural area?

Response: Thank you for the comment. We have now expanded the introduction section to incorporate and address a detailed description of Nepal's healthcare system, the provision of leprosy services, the travel time from referral hospitals, and other contexts to fully appreciate the issues and familiarise the international reader.

“ The healthcare system in Nepal can be categorised into three levels: primary (health posts), secondary (district hospitals), and tertiary (specialised institutions). Leprosy services are decentralised, with health posts serving as the primary point of contact for individuals seeking healthcare in the community. This initial contact serves as the first point for leprosy services such as leprosy diagnosis, MDT initiation, disability grading, complications management, referral and follow-up. The key health care providers responsible for implementing leprosy care and support services in the health post include the health post-in-charge and the leprosy focal person who has either a health assistant or auxiliary health worker background with leprosy training. To guide the healthcare providers, Nepal has standard leprosy operational guidelines adopted and aligned with WHO guidelines to facilitate the implementation of care and support of leprosy for disability prevention and management in health facilities such as health posts, hospitals, and referral centres”

“Despite these efforts, health posts in Rupandehi district lack essential skin smear tests, and skilled laboratory personnel to perform these tests. Consequently, they rely heavily on referrals. Participants from remote areas of Rupandehi district must travel 1-2 hours to reach Butwal for a skin smear test. Thus, healthcare providers refer patients to Butwal for these tests and to specialised hospitals in Kathmandu for leprosy-related complications, such as lepra reaction and ulcer management, if local health facilities cannot address them. Participants from these remote regions face considerable challenges due to transportation difficulties, often needing to travel 10-12 hours to reach specialised hospitals like Anandaban in Kathmandu for managing leprosy complications, resulting in out-of-pocket expenses, long waiting times and inconvenience for socio-economic disadvantaged populations”

# Pages 6 and 7

From lines 114 to 137

Reviewer #1

Background

Your background section emphasizes the importance of implementation fidelity but lacks a clear rationale for studying it in the context of leprosy care in Nepal. To strengthen the justification, briefly explain why exploring implementation fidelity is crucial for improving leprosy care in Nepal. Also, a shorthand definition to implantation fidelity, could be beneficial to the readers unfamiliar to the topic.

Response:

Thank you very much for the feedback. We have expanded the introduction section to incorporate the implementation fidelity definition and rationale for studying in the context of leprosy and improving leprosy care in Nepal.

“ Implementation fidelity refers to the degree to which an intervention or program is delivered as intended by the program developers. High fidelity ensures that established leprosy recommendations and protocols are properly implemented by healthcare providers, resulting in better patient outcomes. In contrast, low fidelity explains why these underperform while being well-designed. Assessing implementation fidelity allows researchers to better understand how and why an intervention works, as well as the extent to which outcomes might be improved”

“ The relevance of studying implementation fidelity in leprosy care and support in the Rupandehi district lies in understanding how effectively Nepal’s leprosy operational guidelines are implemented at the local level. Although national prevalence may be low, persistent leprosy complications, G2D cases, and stigma in leprosy-endemic districts like Rupandehi indicate gaps in policy and pr

---

## [Decision Letter · Decision Letter 1]

Dear Dr. Nepal,

Thank you for submitting your manuscript to PLOS ONE. After careful consideration, we feel that it has merit but does not fully meet PLOS ONE’s publication criteria as it currently stands. Therefore, we invite you to submit a revised version of the manuscript that addresses the points raised during the review process.

We look forward to receiving your revised manuscript.

Kind regards,

Veincent Christian Pepito

Academic Editor

PLOS ONE

Additional Editor Comments:

Dear Author, thank you very much for your revision. Both reviewers have recommended acceptance, but I still have some issues with the paper. Kindly address this so we can move soon to acceptance and publication:

1. Tables 1 and 2, which are respondent profiles, can be transferred to the supplementary appendix. A brief description of who they are, referencing the supplementary appendix, would be sufficient.

2. The reflexivity statement should be rewritten and improved. Reflexivity statement is not just about reflection or critical assessment of biases, etc. It is more about how your backgrounds, beliefs and experiences played a part in the interpretation and development of themes and subthemes. There are many good reflexivity statements from previously published papers; please pattern your statement from it. I want to know what your backgrounds and experiences are and how these have affected the analyses and the generation of themes.

3. I want Supplementary Tables 1 and 2 to be put into the main paper and replace Table 3 as it stands now. I want the comparisons to the guidelines be more explicit in the presentation of Results, similar to the Acta Medica Philippina paper I sent to you in the previous round. You are talking about implementation fidelity so what is done should be compared to what is ideal or in the guidelines. I am fine with the framework used; it just needs to be more harmonized and streamlined. Like, for theme 1 healthcare provider's adherence to leprosy services. There should be a comparison to what is in the guidelines, then how reality is different or similar to it. As it stands, you are only presenting what is currently being done, which does not tell us anything about implementation fidelity.

Reviewers' comments:

Reviewer's Responses to Questions

**Comments to the Author**

Reviewer #1: All comments have been addressed

Reviewer #2: All comments have been addressed

2. Is the manuscript technically sound, and do the data support the conclusions?

Reviewer #1: Yes

Reviewer #2: Yes

3. Has the statistical analysis been performed appropriately and rigorously?

Reviewer #1: Yes

Reviewer #2: N/A

4. Have the authors made all data underlying the findings in their manuscript fully available?

Reviewer #1: Yes

Reviewer #2: Yes

5. Is the manuscript presented in an intelligible fashion and written in standard English?

Reviewer #1: Yes

Reviewer #2: Yes

Reviewer #1: Thank you for addressing and incoporating my feedback. I hope my comments have helped to enhance the quality of your work.

Reviewer #2: I want to confirm that this paper has been well written and can be accepted for publication. NO further comments from me.

**Do you want your identity to be public for this peer review?** For information about this choice, including consent withdrawal, please see our Privacy Policy

Reviewer #1: No

Reviewer #2: No

---

## [Author Response · Author response to Decision Letter 2]

9 Jun 2025

Additional editors comment

Dear Author, thank you very much for your revision. Both reviewers have recommended acceptance, but I still have some issues with the paper. Kindly address this so we can move soon to acceptance and publication.

1. Tables 1 and 2, which are respondent profiles, can be transferred to the supplementary appendix. A brief description of who they are, referencing the supplementary appendix, would be sufficient.

Response:

Thank you so much for the feedback. Initially, we had removed Tables 1 and 2, which were respondent profiles from the main document during 2nd revision submission and transferred them to the supplementary appendix 1 but after the 2nd revision submission to journal we got additional request to include Table 1 and 2 as part of the main manuscript so Table 1 and Table 2 has been included again in the main manuscript.

Additional comments after 2nd revision submission:

“ 1. Please include your Tables 1 and 2 as part of your main manuscript and remove the individual files. Please note that supplementary tables (should remain/ be uploaded) as separate "Supporting Information" files.

Response:

“Thank you for the feedback. Table 1 and Table 2 have been included in the main manuscript, and the individual file has been revised”

#Pages 10-15

From lines 218-233

2. The reflexivity statement should be rewritten and improved. Reflexivity statement is not just about reflection or critical assessment of biases, etc. It is more about how your backgrounds, beliefs and experiences played a part in the interpretation and development of themes and subthemes. There are many good reflexivity statements from previously published papers; please pattern your statement from it. I want to know what your backgrounds and experiences are and how these have affected the analyses and the generation of themes.

Response:

Thank you so much for the constructive feedback. A reflexivity statement has been updated as per the feedback.

“The lead investigator (SN) is a public health professional with over six years of work experience in neglected tropical diseases, including leprosy, at the current study site. SN has completed a Master’s in Public Health (MPH) with a specialisation in Implementation Research (IR). Having witnessed firsthand the barriers to leprosy services, SN approached the research process with extensive knowledge of both policy and ground realities. The data collected in the study were generated in the form of codes and guided by Braun and Clarke’s thematic analysis framework, which influenced the development of themes, including barriers to leprosy care and support and healthcare providers’ adherence to leprosy services.

AP and RAA are a professor and a senior epidemiology and implementation research expert, respectively, with over a decade of expertise in qualitative research. Their specialisation in infectious disease control and health service research contributed significantly to ensuring scientific and methodological rigour. Similarly, AT holds a double degree in MSc. in Public Health and M.A. in Gender Studies; AS is currently pursuing a Master's in Public Health and PCJ completed a Master's in Public Health (MPH). SN, AT, AS and PCJ are familiar with Nepali language and culture and have strong experience in leading qualitative studies on the Nepalese context on a range of public health issues.

All co-authors reviewed the data collection tool and supported the analysis and interpretation of the data. Furthermore, collectively the co-authors agreed on the finalisation of a tool, development of codebook, development of sub-themes and themes, preparation, revision and submission of the manuscript. Throughout the study, all authors approached the study with cultural sensitivity, recognising that their positionality could influence how data is interpreted and analysed.”

This information is updated in the methodology section of the manuscript accordingly.

# pages 17 and 18

From lines 282-304

3. I want Supplementary Tables 1 and 2 to be put into the main paper and replace Table 3 as it stands now. I want the comparisons to the guidelines be more explicit in the presentation of Results, similar to the Acta Medica Philippina paper I sent to you in the previous round. You are talking about implementation fidelity so what is done should be compared to what is ideal or in the guidelines. I am fine with the framework used; it just needs to be more harmonized and streamlined. Like, for theme 1 healthcare provider's adherence to leprosy services. There should be a comparison to what is in the guidelines, then how reality is different or similar to it. As it stands, you are only presenting what is currently being done, which does not tell us anything about implementation fidelity.

Response:

Thank you so much for the feedback.We have now updated the main manuscript with Table 3, which compares the national leprosy operational guideline and actual implementation of leprosy services similar to the Acta Medica Philippina paper.

The codebook, which was previously Table 3 in the main manuscript, has been moved to the Supplementary Appendix 1

#Page 19-20

Lines 320-322

---

## [Editor Report · Decision Letter 2]

Implementation fidelity in leprosy care and support for disability prevention and management in Rupandehi, Nepal: A qualitative study

PONE-D-25-09075R2

Dear Dr. Nepal,

We’re pleased to inform you that your manuscript has been judged scientifically suitable for publication and will be formally accepted for publication once it meets all outstanding technical requirements.

Kind regards,

Veincent Christian Pepito

Academic Editor

PLOS ONE
---

## [Editor Report · Acceptance letter]

PONE-D-25-09075R2

PLOS ONE

Dear Dr. Nepal,

I'm pleased to inform you that your manuscript has been deemed suitable for publication in PLOS ONE. Congratulations! Your manuscript is now being handed over to our production team.

Kind regards,

on behalf of

Mr Veincent Christian Pepito

Academic Editor

PLOS ONE